# ViSoBERT: A Pre-Trained Language Model for Vietnamese Social Media Text Processing

**Quoc-Nam Nguyen[1, 3, *], Thang Chau Phan[1, 3, *], Duc-Vu Nguyen[2, 3], Kiet Van Nguyen[1, 3]**

[1]Faculty of Information Science and Engineering, University of Information Technology,
Ho Chi Minh City, Vietnam

[2]Multimedia Communications Laboratory, University of Information Technology,
Ho Chi Minh City, Vietnam

[3]Vietnam National University, Ho Chi Minh City, Vietnam
{20520644, 20520929}@gm.uit.edu.vn
{vund, kietnv}@uit.edu.vn

## Abstract

English and Chinese, known as resource-rich languages, have witnessed the strong development of transformer-based language models for natural language processing tasks. Although Vietnam has approximately 100M people speaking Vietnamese, several pre-trained models, e.g., PhoBERT, ViBERT, and vELECTRA, performed well on general Vietnamese NLP tasks, including POS tagging and named entity recognition. These pre-trained language models are still limited to Vietnamese social media tasks. In this paper, we present the first monolingual pre-trained language model for Vietnamese social media texts, ViSoBERT, which is pre-trained on a large-scale corpus of high-quality and diverse Vietnamese social media texts using XLM-R architecture. Moreover, we explored our pre-trained model on five important natural language downstream tasks on Vietnamese social media texts: emotion recognition, hate speech detection, sentiment analysis, spam reviews detection, and hate speech spans detection. Our experiments demonstrate that ViSoBERT, with far fewer parameters, surpasses the previous state-of-the-art models on multiple Vietnamese social media tasks. Our ViSoBERT model is available[4] only for research purposes.

**Disclaimer**: This paper contains actual comments on social networks that might be construed as abusive, offensive, or obscene.

## 1 Introduction

Language models based on transformer architecture (Vaswani et al., 2017) pre-trained on large-scale datasets have brought about a paradigm shift in natural language processing (NLP), reshaping how we analyze, understand, and generate text. In particular, BERT (Devlin et al., 2019) and its variants (Liu et al., 2019; Conneau et al., 2020) have achieved state-of-the-art performance on a wide range of downstream NLP tasks, including but not limited to text classification, sentiment analysis, question answering, and machine translation. English is moving for the rapid development of language models across specific domains such as medical (Lee et al., 2019; Rasmy et al., 2021), scientific (Beltagy et al., 2019), legal (Chalkidis et al., 2020), political conflict and violence (Hu et al., 2022), and especially social media (Nguyen et al., 2020; DeLucia et al., 2022; Pérez et al., 2022; Zhang et al., 2022).

Vietnamese is the eighth largest language used over the internet, with around 85 million users across the world[5]. Despite a large amount of Vietnamese data available over the Internet, the advancement of NLP research in Vietnamese is still slow-moving. This can be attributed to several factors, to name a few: the scattered nature of available datasets, limited documentation, and minimal community engagement. Moreover, most existing pre-trained models for Vietnamese were primarily trained on large-scale corpora sourced from general texts (Tran et al., 2020; Nguyen and Tuan Nguyen, 2020; Tran et al., 2023). While these sources provide broad language coverage, they may not fully represent the sociolinguistic phenomena in Vietnamese social media texts. Social media texts often exhibit different linguistic patterns, informal language usage, non-standard vocabulary, lacking diacritics and emoticons that are not prevalent in formal written texts. The limitations of using language models pre-trained on general corpora become apparent when processing Vietnamese social media texts. The models can struggle to accurately un-

---

[*]Equal contribution.

[4]https://huggingface.co/uitnlp/visobert

[5]https://www.internetworldstats.com/stats3.htm

derstand and interpret the informal language, using emoji, teencode, and diacritics used in social media discussions. This can lead to suboptimal performance in Vietnamese social media tasks, including emotion recognition, hate speech detection, sentiment analysis, spam reviews detection, and hate speech spans detection.

We present ViSoBERT, a pre-trained language model designed explicitly for Vietnamese social media texts to address these challenges. ViSoBERT is based on the transformer architecture and trained on a large-scale dataset of Vietnamese posts and comments extracted from well-known social media networks, including Facebook, Tiktok, and Youtube. Our model outperforms existing pre-trained models on various downstream tasks, including emotion recognition, hate speech detection, sentiment analysis, spam reviews detection, and hate speech spans detection, demonstrating its effectiveness in capturing the unique characteristics of Vietnamese social media texts. Our contributions are summarized as follows.

- We presented ViSoBERT, the first PLM based on the XLM-R architecture and pre-training procedure for Vietnamese social media text processing. ViSoBERT is available publicly for research purposes in Vietnamese social media mining. ViSoBERT can be a strong baseline for Vietnamese social media text processing tasks and their applications.

- ViSoBERT produces SOTA performances on multiple Vietnamese downstream social media tasks, thus illustrating the effectiveness of our PLM on Vietnamese social media texts.

- To understand our pre-trained language model deeply, we analyze experimental results on the masking rate, examining social media characteristics, including emojis, teencode, and diacritics, and implementing feature-based extraction for task-specific models.

## 2 Fundamental of Pre-trained Language Models for Social Media Texts

Pre-trained Language Models (PLMs) based on transformers (Vaswani et al., 2017) have become a crucial element in cutting-edge NLP tasks, including text classification and natural language generation. Since then, language models based on transformers related to our study have been reviewed, including PLMs for Vietnamese social media texts.

### 2.1 Pre-trained Language Models for Vietnamese

Several PLMs have recently been developed for processing Vietnamese texts. These models have varied in their architectures, training data, and evaluation metrics. PhoBERT, developed by Nguyen and Tuan Nguyen (2020), is the first general pre-trained language model (PLM) created for the Vietnamese language. The model employs the same architecture as BERT (Devlin et al., 2019) and the same pre-training technique as RoBERTa (Liu et al., 2019) to ensure robust and reliable performance. PhoBERT was trained on a 20GB word-level Vietnamese Wikipedia corpus, which produces SOTA performances on a range of downstream tasks of POS tagging, dependency parsing, NER, and NLI.

Following the success of PhoBERT, viBERT (Tran et al., 2020) and vELECTRA (Tran et al., 2020), both monolingual pre-trained language models based on the BERT and ELECTRA architectures, were introduced. They were trained on substantial datasets, with ViBERT using a 10GB corpus and vELECTRA utilizing an even larger 60GB collection of uncompressed Vietnamese text. viBERT4news[6] was published by NlpHUST, a Vietnamese version of BERT trained on more than 20 GB of news datasets. For Vietnamese text summarization, BARTpho (Tran et al., 2022) is presented as the first large-scale monolingual seq2seq models pre-trained for Vietnamese, based on the seq2seq denoising autoencoder BART. Moreover, ViT5 (Phan et al., 2022) follows the encoder-decoder architecture proposed by Vaswani et al. (2017) and the T5 framework proposed by Raffel et al. (2020). Many language models are designed for general use, while the availability of strong baseline models for domain-specific applications remains limited. Since then, Minh et al. (2022) introduced ViHealthBERT, the first domain-specific PLM for Vietnamese healthcare.

### 2.2 Pre-trained Language Models for Social Media Texts

Multiple PLMs were introduced for social media for multilingual and monolinguals. BERTweet (Nguyen et al., 2020) was presented as the first public large-scale PLM for English Tweets. BERTweet has the same architecture as BERT$_{Base}$ (Devlin et al., 2019) and is trained using the RoBERTa pre-

---

[6]https://github.com/bino282/bert4news

training procedure (Liu et al., 2019). Koto et al. (2021) proposed IndoBERTweet, the first large-scale pre-trained model for Indonesian Twitter. IndoBERTweet is trained by extending a monolingually trained Indonesian BERT model with an additive domain-specific vocabulary. RoBERTuito, presented in Pérez et al. (2022), is a robust transformer model trained on 500 million Spanish tweets. RoBERTuito excels in various language contexts, including multilingual and code-switching scenarios, such as Spanish and English. TWilBert (Ángel González et al., 2021) is proposed as a specialization of BERT architecture both for the Spanish language and the Twitter domain to address text classification tasks in Spanish Twitter.

Bernice, introduced by DeLucia et al. (2022), is the first multilingual pre-trained encoder designed exclusively for Twitter data. This model uses a customized tokenizer trained solely on Twitter data and incorporates a larger volume of Twitter data (2.5B tweets) than most BERT-style models. Zhang et al. (2022) introduced TwHIN-BERT, a multilingual language model trained on 7 billion Twitter tweets in more than 100 different languages. It is designed to handle short, noisy, user-generated text effectively. Previously, (Barbieri et al., 2022) extended the training of the XLM-R (Conneau et al., 2020) checkpoint using a data set comprising 198 million multilingual tweets. As a result, XLM-T is adapted to the Twitter domain and was not exclusively trained on data from within that domain.

## 3   ViSoBERT

This section presents the architecture, pre-training data, and our custom tokenizer on Vietnamese social media texts for ViSoBERT.

### 3.1   Pre-training Data

We crawled textual data from Vietnamese public social networks such as Facebook[7], Tiktok[8], and YouTube[9] which are the three most well known social networks in Vietnam, with 52.65, 49.86, and 63.00 million users[10], respectively, in early 2023.

To effectively gather data from these platforms, we harnessed the capabilities of specialized tools provided by each platform.

1. **Facebook**:  We crawled comments from Vietnamese-verified pages by Facebook posts via the Facebook Graph API[11] between January 2016 and December 2022.

2. **TikTok**:  We collected comments from Vietnamese-verified channels by TikTok through TikTok Research API[12] between January 2020 and December 2022.

3. **YouTube**:  We scrapped comments from Vietnam-verified channels' videos by YouTube via YouTube Data API[13] between January 2016 and December 2022.

**Pre-processing Data:** Pre-processing is vital for models consuming social media data, which is massively noisy, and has user handles (@username), hashtags, emojis, misspellings, hyperlinks, and other noncanonical texts. We perform the following steps to clean the dataset: removing noncanonical texts, removing comments including links, removing excessively repeated spam and meaningless comments, removing comments including only user handles (@username), and keeping emojis in training data.

As a result, our pretraining data after crawling and preprocessing contains 1GB of uncompressed text. Our pretraining data is available only for research purposes.

### 3.2   Model Architecture

Transformers (Vaswani et al., 2017) have significantly advanced NLP research using trained models in recent years. Although language models (Nguyen and Tuan Nguyen, 2020; Nguyen and Nguyen, 2021) have also proven effective on a range of Vietnamese NLP tasks, their results on Vietnamese social media tasks (Nguyen et al., 2022) need to be significantly improved. To address this issue, taking into account successful hyperparameters from XLM-R (Conneau et al., 2020), we proposed ViSoBERT, a transformer-based model in the style of XLM-R architecture with 768 hidden units, 12 self-attention layers, and 12 attention heads, and used a masked language objective (the same as Conneau et al. (2020)).

---

[7]https://www.facebook.com/

[8]https://www.tiktok.com/

[9]https://www.youtube.com/

[10]https://datareportal.com/reports/digital-2023-vietnam

[11]https://developers.facebook.com/

[12]https://developers.tiktok.com/products/research-api/

[13]https://developers.google.com/youtube/v3

### 3.3 The Vietnamese Social Media Tokenizer

To the best of our knowledge, ViSoBERT is the first PLM with a custom tokenizer for Vietnamese social media texts. Bernice (DeLucia et al., 2022) was the first multilingual model trained from scratch on Twitter[14] data with a custom tokenizer; however, Bernice's tokenizer doesn't handle Vietnamese social media text effectively. Moreover, existing Vietnamese pre-trained models' tokenizers perform poorly on social media text because of different domain data training. Therefore, we developed the first custom tokenizer for Vietnamese social media texts.

Owing to the ability to handle raw texts of SentencePiece (Kudo and Richardson, 2018) without any loss compared to Byte-Pair Encoding (Conneau et al., 2020), we built a custom tokenizer on Vietnamese social media by SentencePiece on the whole training dataset. A model has better coverage of data than another when *fewer* subwords are needed to represent the text, and the subwords are *longer* (DeLucia et al., 2022). Figure 2 (in Appendix A) displays the mean token length for each considered model and group of tasks. ViSoBERT achieves the shortest representations for all Vietnamese social media downstream tasks compared to other PLMs.

Emojis and teencode are essential to the "language" on Vietnamese social media platforms. Our custom tokenizer's capability to decode emojis and teencode ensure that their semantic meaning and contextual significance are accurately captured and incorporated into the language representation, thus enhancing the overall quality and comprehensiveness of text analysis and understanding.

To assess the tokenized ability of Vietnamese social media textual data, we conducted an analysis of several data samples. Table 1 shows several actual social comments and their tokenizations with the tokenizers of the two pre-trained language models, ViSoBERT and PhoBERT, the best strong baseline. The results show that our custom tokenizer performed better compared to others.

## 4 Experiments and Results

### 4.1 Experimental Settings

We accumulate gradients over one step to simulate a batch size of 128. When pretraining from scratch, we train the model for 1.2M steps in 12 epochs. We trained our model for about three days on 2×RTX4090 GPUs (24GB). Each sentence is tokenized and masked dynamically with a probability equal to 30% (which is extensively experimented on Section 5.1 to explore the optimal value). Further details on hyperparameters and training can be found in Table 6 of Appendix B.

**Downstream tasks.** To evaluate ViSoBERT, we used five Vietnamese social media datasets available for research purposes, as summarized in Table 2. The downstream tasks include emotion recognition (UIT-VSMEC) (Ho et al., 2020), hate speech detection (UIT-ViHSD) (Luu et al., 2021), sentiment analysis (SA-VLSP2016) (Nguyen et al., 2018), spam reviews detection (ViSpamReviews) (Dinh et al., 2022), and hate speech spans detection (UIT-ViHOS) (Hoang et al., 2023).

**Fine-tuning.** We conducted empirical fine-tuning for all pre-trained language models using the *simpletransformers*[15]. Our fine-tuning process followed standard procedures, most of which are outlined in (Devlin et al., 2019). For all tasks mentioned above, we use a batch size of 40, a maximum token length of 128, a learning rate of 2e-5, and AdamW optimizer (Loshchilov and Hutter, 2019) with an epsilon of 1e-8. We executed a 10-epoch training process and evaluated downstream tasks using the best-performing model from those epochs. Furthermore, none of the pre-processing techniques is applied in all datasets to evaluate our PLM's ability to handle raw texts.

**Baseline models.** To establish the main baseline models, we utilized several well-known PLMs, including monolingual and multilingual, to support Vietnamese NLP social media tasks. The details of each model are shown in Table 3.

- **Monolingual language models**: viBERT (Tran et al., 2020) and vELECTRA (Tran et al., 2020) are PLMs for Vietnamese based on BERT and ELECTRA architecture, respectively. PhoBERT, which is based on BERT architecture and RoBERTa pre-training procedure, (Nguyen and Tuan Nguyen, 2020) is the first large-scale monolingual language model pre-trained for Vietnamese; PhoBERT obtains state-of-the-art performances on a range of Vietnamese NLP tasks.

- **Multilingual language models**: Additionally, we incorporated two multilingual PLMs,

---

[14]https://twitter.com/

[15]https://simpletransformers.ai/ (ver 0.63.11)

| Comments | ViSoBERT | PhoBERT |
|---|---|---|
| concặc cáilồn gìđây🙂🙂🙂 
 *English*: Wut is dis fuckingd1ck 🙂🙂🙂 | \<s\>, "conc", "ặc", "cái", "l", "ồn", "gì", "đây", "🙂🙂🙂", \</s\> | \<s\>, "c o n @ @", "c @ @", "ặ c", "c á @ @", "i l @ @", "ồ n", "g @ @","ì @ @", "đ â y", \<unk\>, \<unk\>, \<unk\>, \</s\> |
| e cảmơn anh😎😎 
 *English*: Thankyou 😎😎 | \<s\>, "e", "cảm", "ơn", "anh", "😎", "😎", \</s\> | \<s\>, "e", "c ả @ @", "m @ @", "ơ n", "a n h", \<unk\>, \<unk\>, \</s\> |
| d4y l4 vj du cko mot cau teencode 
 *English*: Th1s 1s 4 teencode s3nt3nc3 | \<s\>, "d", "4", "y", "l", "4", "vj", "du", "cko", "mot", "cau", "teen", "code", \</s\> | \<s\>, "d @ @", "4 @ @", "y", "l @ @", "4", "v @ @", "j", "d u", "c k @ @", "o", "m o @ @", "t", "c a u"; "t e @ @", "e n @ @", "c o d e", \</s\> |

Table 1: Actual social comments and their tokenizations with the tokenizers of the two pre-trained language models, ViSoBERT and PhoBERT.

| Dataset | Train | Dev | Test | Task | Evaluation Metrics | Classes |
|---|---|---|---|---|---|---|
| UIT-VSMEC | 5,548 | 686 | 693 | Emotion Recognition (ER) | | 7 |
| UIT-HSD | 24,048 | 2,672 | 6,680 | Hate Speech Detection (HSD) | | 3 |
| SA-VLSP2016 | 5,100 | - | 1,050 | Sentiment Analysis (SA) | Acc, WF1, MF1 (%) | 3 |
| ViSpamReviews | 14,306 | 1,590 | 3,974 | Spam Reviews Detection (SRD) | | 4 |
| ViHOS | 8,844 | 1,106 | 1,106 | Hate Speech Spans Detection (HSSD) | | 3 |

Table 2: Statistics and descriptions of Vietnamese social media processing tasks. Acc, WF1, and MF1 denoted Accuracy, weighted F1-score, and macro F1-score metrics, respectively.

mBERT (Devlin et al., 2019) and XLM-R (Conneau et al., 2020), which were previously shown to have competitive performances to monolingual Vietnamese models. XLM-R, a cross-lingual PLM introduced by Conneau et al. (2020), has been trained in 100 languages, among them Vietnamese, utilizing a vast 2.5TB Clean CommonCrawl dataset. XLM-R presents notable improvements in various downstream tasks, surpassing the performance of previously released multilingual models such as mBERT (Devlin et al., 2019) and XLM (Lample and Conneau, 2019).

- **Multilingual social media language models:** To ensure a fair comparison with our PLM, we integrated multiple multilingual social media PLMs, including XLM-T (Barbieri et al., 2022), TwHIN-BERT (Zhang et al., 2022), and Bernice (DeLucia et al., 2022).

## 4.2 Main Results

Table 4 shows ViSoBERT's scores with the previous highest reported results on other PLMs using the same experimental setup. It is clear that our ViSoBERT produces new SOTA performance results for multiple Vietnamese downstream social media tasks without any pre-processing technique.

**Emotion Recognition Task**: PhoBERT and TwHIN-BERT archive the previous SOTA performances on monolingual and multilingual models, respectively. ViSoBERT obtains 68.10%, 68.37%, and 65.88% of Acc, WF1, and MF1, respectively, significantly higher than these PhoBERT and TwHIN-BERT models.

**Hate Speech Detection Task**: ViSoBERT achieves significant improvements over previous state-of-the-art models, PhoBERT and TwHIN-BERT, with scores of 88.51%, 88.31%, and 68.77% in Acc, WF1, and MF1, respectively. Notably, these achievements are made despite the presence of bias within the dataset[16].

**Sentiment Analysis Task**: XLM-R archived SOTA performance on three evaluation metrics. However, there is no significant increase in performance on this downstream task, for 0.45%, 0.46%, and 0.46% higher on Acc, WF1, and MF1 compared to our pre-trained language model, PhoBERT_{Large}. The SA-VLSP2016 dataset domain is technical article reviews, including TinhTe[17] and VnExpress[18], which are often used as Vietnamese standard data. The reviews or comments in these newspapers are in proper form. While most of the dataset is sourced from articles, it also includes data from Facebook[19], a Vietnamese social media platform that accounts for only 12.21% of the dataset. Therefore, the dataset does not fully capture Vietnamese social media platforms' diverse characteristics and infor-

---

[16]UIT-HSD is massively imbalanced, included 19,886; 1,606; and 2,556 of CLEAN, OFFENSIVE, and HATE class.
[17]https://tinhte.vn/
[18]https://vnexpress.net/
[19]https://www.facebook.com/

| Model | #Layers | #Heads | #Steps | #Batch | Domain Data | #Params | #Vocab | #MSL | CSMT |
|---|---|---|---|---|---|---|---|---|---|
| viBERT (Tran et al., 2020) | 12 | 12 | - | 16 | Vietnamese News | - | 30K | 256 | No |
| vELECTRA (Tran et al., 2020) | 12 | 3 | - | 16 | NewsCorpus + OscarCorpus | - | 30K | 256 | No |
| PhoBERT$_{Base}$ (Nguyen and Tuan Nguyen, 2020) | 12 | 12 | 540K | 1024 | ViWiki + ViNews | 135M | 64K | 256 | No |
| PhoBERT$_{Large}$ (Nguyen and Tuan Nguyen, 2020) | 24 | 16 | 1.08M | 512 | ViWiki + ViNews | 370M | 64K | 256 | No |
| mBERT (Devlin et al., 2019) | 12 | 12 | 1M | 256 | BookCorpus + EnWiki | 110M | 30K | 512 | No |
| XLM-R$_{Base}$ (Conneau et al., 2020) | 12 | 12 | 1.5M | 8192 | CommonCrawl + Wiki | 270M | 250K | 512 | No |
| XLM-R$_{Large}$ (Conneau et al., 2020) | 24 | 16 | 1.5M | 8192 | CommonCrawl + Wiki | 550M | 250K | 512 | No |
| XLM-T (Barbieri et al., 2022) | 12 | 12 | - | 8192 | Multilingual Tweets | - | 250k | 512 | No |
| TwHIN-BERT$_{Base}$ (Zhang et al., 2022) | 12 | 12 | 500K | 6K | Multilingual Tweets | 135M to 278M | 250K | 128 | No |
| TwHIN-BERT$_{Large}$ (Zhang et al., 2022) | 24 | 16 | 500K | 8K | Multilingual Tweets | 550M | 250K | 128 | No |
| Bernice (DeLucia et al., 2022) | 12 | 12 | 405K+ | 8192 | Multilingual Tweets | 270M | 250K | 128 | Yes |
| ViSoBERT (Ours) | 12 | 12 | 1.2M | 128 | Vietnamese social media | 97M | 15K | 512 | Yes |

Table 3: Detailed information about baselines and our PLM. #Layers, #Heads, #Batch, #Params, #Vocab, #MSL, and CSMT indicate the number of hidden layers, number of attention heads, batch size value, domain training data, number of total parameters, vocabulary size, max sequence length, and custom social media tokenizer, respectively.

| Model | Avg | Emotion Regcognition | | | Hate Speech Detection | | | Sentiment Analysis | | | Spam Reviews Detection | | | Hate Speech Spans Detection | | |
|---|---|---|---|---|---|---|---|---|---|---|---|---|---|---|---|---|
| | | Acc | WF1 | MF1 | Acc | WF1 | MF1 | Acc | WF1 | MF1 | Acc | WF1 | MF1 | Acc | WF1 | MF1 |
| viBERT | 71.57 | 61.91 | 61.98 | 59.70 | 85.34 | 85.01 | 62.07 | 74.85 | 74.73 | 74.73 | 89.93 | 89.79 | 76.80 | 90.42 | 90.45 | 84.55 |
| vELECTRA | 72.43 | 64.79 | 64.71 | 61.95 | 86.96 | 86.37 | 63.95 | 74.95 | 74.88 | 74.88 | 89.83 | 89.68 | 76.23 | 90.59 | 90.58 | 85.12 |
| PhoBERT$_{Base}$ | 72.81 | 63.49 | 63.36 | 61.41 | 87.12 | 86.81 | 65.01 | 75.72 | 75.52 | 75.52 | 89.83 | 89.75 | 76.18 | 91.32 | 91.38 | 85.92 |
| PhoBERT$_{Large}$ | 73.47 | 64.71 | 64.66 | 62.55 | 87.32 | 86.98 | 65.14 | 76.52 | 76.36 | 76.22 | 90.12 | 90.03 | 76.88 | 91.44 | 91.46 | 86.56 |
| mBERT (cased) | 68.07 | 56.27 | 56.17 | 53.48 | 83.55 | 83.99 | 60.62 | 67.14 | 67.16 | 67.16 | 89.05 | 88.89 | 74.52 | 89.88 | 89.87 | 84.57 |
| mBERT (uncased) | 67.66 | 56.23 | 56.11 | 53.32 | 83.38 | 81.27 | 58.92 | 67.25 | 67.22 | 67.22 | 88.92 | 88.72 | 74.32 | 89.84 | 89.82 | 84.51 |
| XLM-R$_{Base}$ | 72.08 | 60.92 | 61.02 | 58.67 | 86.36 | 86.08 | 63.39 | 76.38 | 76.38 | 76.38 | 90.16 | 89.96 | 76.55 | 90.74 | 90.72 | 85.42 |
| XLM-R$_{Large}$ | 73.40 | 62.44 | 61.37 | 60.25 | 87.15 | 86.86 | 65.13 | **78.28** | **78.21** | **78.21** | 90.36 | 90.31 | 76.75 | 91.52 | 91.50 | 86.66 |
| XLM-T | 72.23 | 64.64 | 64.37 | 59.86 | 86.22 | 86.12 | 63.48 | 75.66 | 75.60 | 75.60 | 90.07 | 90.11 | 76.66 | 90.88 | 90.88 | 85.53 |
| TwHIN-BERT$_{Base}$ | 71.60 | 61.49 | 60.88 | 57.97 | 86.63 | 86.23 | 63.67 | 73.76 | 73.72 | 73.72 | 90.25 | 90.35 | 76.98 | 90.99 | 90.90 | 85.67 |
| TwHIN-BERT$_{Large}$ | 73.42 | 64.21 | 64.29 | 61.12 | 87.23 | 86.78 | 65.23 | 76.92 | 76.83 | 76.83 | 90.47 | 90.42 | 77.28 | 91.45 | 91.47 | 86.65 |
| Bernice | 72.49 | 64.21 | 64.27 | 60.68 | 86.12 | 86.48 | 64.32 | 74.57 | 74.90 | 74.90 | 90.22 | 90.21 | 76.89 | 90.48 | 90.06 | 85.67 |
| ViSoBERT | **75.65** | **68.10** | **68.37** | **65.88**‡ | **88.51** | **88.31** | **68.77**‡ | 77.83 | 77.75 | 77.75 | **90.99** | **90.92** | **79.06**‡ | **91.62** | **91.57** | **86.80** |

Table 4: Performances on downstream Vietnamese social media tasks of previous state-of-the-art monolingual and multilingual PLMs without pre-processing techniques. Avg denoted the average MF1 score of each PLM. ‡ denotes that the highest result is statistically significant at $p < 0.01$ compared to the second best, using a paired t-test.

mal language. However, ViSoBERT still surpassed other baselines by obtaining 1.31%/0.91% Acc, 1.39%/0.92% WF1, and 1.53%/0.92% MF1 compared to PhoBERT/TwHIN-BERT.

**Spam Reviews Detection Task**: ViSoBERT performed better than the top two baseline models, PhoBERT and TwHIN-BERT. Specifically, it achieved 0.8%, 0.9%, and 2.18% higher scores in accuracy (Acc), weighted F1 (WF1), and micro F1 (MF1) compared to PhoBERT. When compared to TwHIN-BERT, ViSoBERT outperformed it with 0.52%, 0.50%, and 1.78% higher scores in Acc, WF1, and MF1, respectively.

**Hate Speech Spans Detection Task**[20]: Our pre-trained ViSoBERT boosted the results up to 91.62%, 91.57%, and 86.80% on Acc, WF1, and MF1, respectively. While the difference is insignificant, ViSoBERT indicates an outstanding ability to capture Vietnamese social media information compared to other PLMs (see Section 5.3).

**Multilingual social media PLMs**: The results show that ViSoBERT consistently outperforms

---

[20]For the Hate Speech Spans Detection task, we evaluate the total of spans on each comment rather than spans of each word in Hoang et al. (2023) to retain the context of each comment.

XLM-T and Bernice in five Vietnamese social media tasks. It's worth noting that XLM-T, TwHIN-BERT, and Bernice were all exclusively trained on data from the Twitter platform. However, this approach has limitations when applied to the Vietnamese context. The training data from this source may not capture the intricate linguistic and contextual nuances prevalent in Vietnamese social media because Twitter is not widely used in Vietnam.

## 5 Result Analysis and Discussion

In this section, we consider the improvement of our PLM more compared to powerful others, including PhoBERT and TwHIN-BERT, in terms of different aspects. Firstly, we investigate the effects of masking rate on our pre-trained model performance (see Section 5.1). Additionally, we examine the influence of social media characteristics on the model's ability to process and understand the language used in these social contexts (see Section 5.2). Lastly, we employed feature-based extraction techniques on task-specific models to verify the potential of leveraging social media textual data to enhance word representations (see Section 5.3).

### 5.1 Impact of Masking Rate on Vietnamese Social Media PLM

For the first time presenting the Masked Language Model, Devlin et al. (2019) consciously utilized a random masking rate of 15%. The authors believed masking too many tokens could lead to losing crucial contextual information required to decode them accurately. Additionally, the authors felt that masking too few tokens would harm the training process and make it less effective. However, according to Wettig et al. (2023), 15% is not universally optimal for model and training data.

We experiment with masking rates ranging from 10% to 50% and evaluate the model's performance on five downstream Vietnamese social media tasks. Figure 1 illustrates the results obtained from our experiments with six different masking rates. Interestingly, our pre-trained ViSoBERT achieved the highest performance when using a masking rate of 30%. This suggests a delicate balance between the amount of contextual information retained and the efficiency of the training process, and an optimal masking rate can be found within this range.

However, the optimal masking rate also depends on the specific task. For instance, in the hate speech detection task, we found that a masking rate of 50% yielded the best results, surpassing other masking rate values. This implies that the optimal masking rate may vary depending on the nature and requirements of different tasks.

Considering the overall performance across multiple tasks, we determined that a masking rate of 30% produced the optimal balance for our pre-trained ViSoBERT model. Consequently, we adopted this masking rate for ViSoBERT, ensuring efficient and effective utilization of contextual information during training.

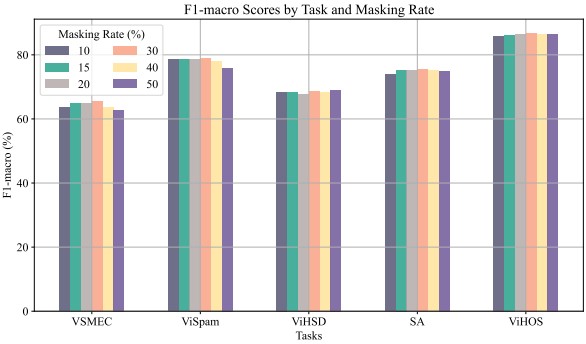

Figure 1: Impact of masking rate on our pre-trained ViSoBERT in terms of MF1.

### 5.2 Impact of Vietnamese Social Media Characteristics

Emojis, teencode, and diacritics are essential features of social media, especially Vietnamese social media. The ability of the tokenizer to decode emojis and the ability of the model to understand the context of teencode and diacritics are crucial. Hence, to evaluate the performance of ViSoBERT on social media characteristics, comprehensive experiments were conducted among several strong PLMs: PM4ViSMT, PhoBERT, and TwHIN-BERT.

**Impact of Emoji on PLMs:** We conducted two experimental procedures to comprehensively investigate the importance of emojis, including converting emojis to general text and removing emojis.

Table 5 shows our detailed setting and experimental results on downstream tasks and pre-trained models. The results indicate a moderate reduction in performance across all downstream tasks when emojis are removed or converted to text in our pre-trained ViSoBERT model. Our pre-trained decreases 0.62% Acc, 0.55% WF1, and 0.78% MF1 on Average for downstream tasks while converting emojis to text. In addition, an average reduction of 1.33% Acc, 1.32% WF1, and 1.42% MF1 can be seen in our pre-trained model while removing all emojis in each comment. This is because when emojis are converted to text, the context of the comment is preserved, while removing all emojis results in the loss of that context.

This trend is also observed in the TwHIN-BERT model, specifically designed for social media processing. However, TwHIN-BERT slightly improves emotion recognition and spam reviews detection tasks compared to its competitors when operating on raw texts. Nevertheless, this improvement is marginal and insignificant, as indicated by the small increments of 0.61%, 0.13%, and 0.21% in Acc, WF1, and MF1 on the emotion recognition task, respectively, and 0.08% Acc, 0.05% WF1, and 0.04% MF1 on spam reviews detection task. One potential reason for this phenomenon is that TwHIN-BERT and ViSoBERT are PLMs trained on emojis datasets. Consequently, these models can comprehend the contextual meaning conveyed by emojis. This finding underscores the importance of emojis in social media texts.

In contrast, there is a general trend of improved performance across a range of downstream tasks when removing or converting emojis to text on

| Model | Emotion Regconition | | | Hate Speech Detection | | | Sentiment Analysis | | | Spam Reviews Detection | | | Hate Speech Spans Detection | | |
|---|---|---|---|---|---|---|---|---|---|---|---|---|---|---|---|
| | Acc | WF1 | MF1 | Acc | WF1 | MF1 | Acc | WF1 | MF1 | Acc | WF1 | MF1 | Acc | WF1 | MF1 |
| *Converting emojis to text* | | | | | | | | | | | | | | | |
| PhoBERT$_{Large}$ | 66.08 | 66.15 | 63.35 | 87.43 | 87.22 | 65.32 | 76.73 | 76.48 | 76.48 | 90.35 | 90.11 | 77.02 | 92.16 | 91.98 | 86.72 |
| Δ | ↑ 1.37 | ↑ 1.49 | ↑ 0.80 | ↑ 0.11 | ↑ 0.24 | ↑ 0.18 | ↓ 0.21 | ↓ 0.12 | ↓ 0.12 | ↑ 0.23 | ↑ 0.08 | ↑ 0.14 | ↑ 0.72 | ↑ 0.52 | ↑ 0.16 |
| TwHIN-BERT$_{Large}$ | 64.82 | 64.42 | 61.33 | 86.03 | 85.52 | 63.52 | 75.42 | 75.95 | 75.95 | 90.55 | 90.47 | 77.32 | 92.21 | 92.01 | 86.84 |
| Δ | ↑ 0.61 | ↑ 0.13 | ↑ 0.21 | ↓ 1.20 | ↓ 1.26 | ↓ 1.71 | ↓ 1.50 | ↓ 0.88 | ↓ 0.88 | ↑ 0.08 | ↑ 0.05 | ↑ 0.04 | ↑ 0.76 | ↑ 0.54 | ↑ 0.19 |
| ViSoBERT [♣] | 67.53 | 67.93 | 65.42 | 87.82 | 87.88 | 67.25 | 76.95 | 76.85 | 76.85 | 90.22 | 90.18 | 78.25 | 92.42 | 92.11 | 87.01 |
| Δ | ↓ 0.57 | ↓ 0.44 | ↓ 0.46 | ↓ 0.69 | ↓ 0.41 | ↓ 1.49 | ↓ 0.88 | ↓ 0.90 | ↓ 0.90 | ↓ 0.77 | ↓ 0.74 | ↓ 0.81 | ↑ 0.80 | ↑ 0.54 | ↑ 0.21 |
| *Removing emojis* | | | | | | | | | | | | | | | |
| PhoBERT$_{Large}$ | 65.21 | 65.14 | 62.81 | 87.25 | 86.72 | 64.85 | 76.72 | 76.48 | 76.48 | 90.21 | 90.09 | 77.02 | 91.53 | 91.51 | 86.62 |
| Δ | ↑ 0.50 | ↑ 0.48 | ↑ 0.26 | ↓ 0.07 | ↓ 0.26 | ↓ 0.29 | ↑ 0.20 | ↑ 0.12 | ↑ 0.12 | ↑ 0.09 | ↑ 0.06 | ↑ 0.10 | ↑ 0.09 | ↑ 0.05 | ↑ 0.09 |
| TwHIN-BERT$_{Large}$ | 62.03 | 62.14 | 59.25 | 86.98 | 86.32 | 64.22 | 75.00 | 75.11 | 75.11 | 89.83 | 89.75 | 76.85 | 91.32 | 91.33 | 86.42 |
| Δ | ↓ 2.18 | ↓ 1.15 | ↓ 1.87 | ↓ 0.25 | ↓ 0.46 | ↓ 1.01 | ↓ 1.92 | ↓ 1.72 | ↓ 1.72 | ↓ 0.64 | ↓ 0.67 | ↓ 0.43 | ↓ 0.13 | ↓ 0.14 | ↓ 0.23 |
| ViSoBERT [♦] | 66.52 | 67.02 | 64.55 | 87.32 | 87.12 | 66.98 | 76.25 | 75.98 | 75.98 | 89.72 | 89.69 | 77.95 | 91.58 | 91.53 | 86.72 |
| Δ | ↓ 1.58 | ↓ 1.35 | ↓ 1.33 | ↓ 1.19 | ↓ 1.19 | ↓ 1.79 | ↓ 1.58 | ↓ 1.77 | ↓ 1.77 | ↓ 1.27 | ↓ 1.23 | ↓ 1.11 | ↓ 0.04 | ↓ 0.04 | ↓ 0.08 |
| ViSoBERT [♠] | **68.10** | **68.37** | **65.88** | **88.51** | **88.31** | **68.77** | **77.83** | **77.75** | **77.75** | **90.99** | **90.92** | **79.06** | **91.62** | **91.57** | **86.80** |

Table 5: Performances of pre-trained models on downstream Vietnamese social media tasks by applying two emojis pre-processing techniques. [♣], [♦], and [♠] denoted our pre-trained language model ViSoBERT converting emoji to text, removing emojis and without applying any pre-processing techniques, respectively. Δ denoted the increase (↑) and the decrease (↓) in performances of the PLMs compared to their competitors without applying any pre-processing techniques.

PhoBERT, the Vietnamese SOTA pre-trained language model. PhoBERT is a PLM on a general text (Vietnamese Wikipedia) dataset containing no emojis; therefore, when PhoBERT encounters an emoji, it treats it as an unknown token (see Table 1 Appendix B). Therefore, while applying emoji pre-processing techniques, including converting emoijs to text and removing emojis, PhoBERT produces better performances compared to raw text.

Our pre-trained model ViSoBERT on raw texts outperformed PhoBERT and TwHIN-BERT even when applying two pre-processing emojis techniques. This claims our pre-trained model's ability to handle Vietnamese social media raw texts.

**Impact of Teencode on PLMs:** Due to informal and casual communication, social media texts often lead to common linguistic errors, such as misspellings and teencode. For example, the phrase "ăng kơmmmmm" should be "ăn cơm" ("Eat rice" in English), and "ko" should be "không" ("No" in English). To address this challenge, Nguyen and Van Nguyen (2020) presented several rules to standardize social media texts. Building upon the previous work, Quoc Tran et al. (2023) proposed a strict and efficient pre-processing technique to clean comments on Vietnamese social media.

Table 7 (in Appendix C) shows the results with and without standardizing teencode on social media texts. There is an uptrend across PhoBERT, TwHIN-BERT, and ViSoBERT while applying standardized pre-processing techniques. ViSoBERT, with standardized pre-processing techniques, outperforms almost downstream tasks but spam reviews detection. The possible reason is that

the ViSpamReviews dataset contains samples in which users use the word with duplicated characters to improve the comment length while standardizing teencodes leads to misunderstanding.

Experimental results strongly suggest that the improvement achieved by applying complex pre-processing techniques to pre-trained models in the context of Vietnamese social media text is relatively insignificant. Despite the considerable time and effort invested in designing and implementing these techniques, the actual gains in PLMs performance are not substantial and unstable.

**Impact of Vietnamese Diacritics on PLMs:** Vietnamese words are created from 29 letters, including seven letters using four diacritics (ă, â-ê-ô, ơ-ư, and đ) and five diacritics used to designate tone (as in à, á, ả, ã, and ạ) (Ngo, 2020). These diacritics create meaningful words by combining syllables (Le-Hong, 2021). For instance, the syllable "ngu" can be combined with five different diacritic marks, resulting in five distinct syllables: "ngú", "ngù", "ngụ", "ngủ", and "ngũ". Each of these syllables functions as a standalone word.

However, social media text does not always adhere to proper writing conventions. Due to various reasons, many users write text without diacritic marks when commenting on social media platforms. Consequently, effectively handling diacritics in Vietnamese social media becomes a critical challenge. To evaluate the PLMs' capability to address this challenge, we experimented by removing all diacritic marks from the datasets of five downstream tasks. This experiment aimed to assess the model's performance in processing text without

diacritics and determine its ability to understand Vietnamese social media content in such cases.

Table 8 (in Appendix C) presents the results of the two best baselines compared to our pre-trained diacritics experiments. The experimental results reveal that the performance of all pre-trained models, including ours, exhibited a significant decrease when dealing with social media comments lacking diacritics. This decline in performance can be attributed to the loss of contextual information caused by the removal of diacritics. The lower the percentage of diacritic removal in each comment, the more significant the performance improvement in all PLMs. However, our ViSoBERT demonstrated a relatively minor reduction in performance across all downstream tasks. This suggests that our model possesses a certain level of robustness and adaptability in comprehending and analyzing Vietnamese social media content without diacritics. We attribute this to the efficiency of the in-domain pre-training data of ViSoBERT.

In contrast, PhoBERT and TwHIN-BERT experienced a substantial drop in performance across the benchmark datasets. These PLMs struggled to cope with the absence of diacritics in Vietnamese social media comments. The main reason is that the tokenizer of PhoBERT can not encode non-diacritics comments due to not including those in pre-training data. Several tokenized examples of the three best PLMs are presented in Table 10 (in Appendix F). Thus, the significant decrease in its performance highlights the challenge of handling diacritics on Vietnamese social media. While handling diacritics remains challenging, ViSoBERT demonstrates promising performance, suggesting the potential for specialized language models tailored for Vietnamese social media analysis.

### 5.3 Impact of Feature-based Extraction to Task-Specific Models

In task-specific models, the contextualized word embeddings from PLMs are typically employed as input features. We aim to assess the quality of contextualized word embeddings generated by PhoBERT, TwHIN-BERT, and ViSoBERT to verify whether social media data can enhance word representation. These contextualized word embeddings are applied as embedding features to BiLSTM, and BiGRU is randomly initialized before the classification layer. We append a linear prediction layer to the last transformer layer of each PLM regard-

ing the first subword of each word token, which is similar to Devlin et al. (2019).

Our experiment results (see Table 9 in Appendix C) demonstrate that the word embeddings generated by our pre-trained language model ViSoBERT outperform other pre-trained embeddings when utilized with BiLSTM and BiGRU for all downstream tasks. The experimental results indicate the significant impact of leveraging social media text data for enriching word embeddings. Furthermore, this finding underscores the effectiveness of our model in capturing the linguistic characteristics prevalent in Vietnamese social media texts.

Figure 3 (in Appendix D) presents the performances of the PLMs as input features to BiLSTM and BiGRU on the dev set per epoch in terms of MF1. The results demonstrate that ViSoBERT reaches its peak MF1 score in only 1 to 3 epochs, whereas other PLMs typically require an average of 8 to 10 epochs to achieve on-par performance. This suggests that ViSoBERT has a superior capability to extract Vietnamese social media information compared to other models.

## 6 Conclusion and Future Work

We presented ViSoBERT, a novel large-scale monolingual pre-trained language model on Vietnamese social media texts. We illustrated that ViSoBERT with fewer parameters outperforms recent strong pre-trained language models such as viBERT, vELECTRA, PhoBERT, XLM-R, XLM-T, TwHIN-BERT, and Bernice and achieves state-of-the-art performances for multiple downstream Vietnamese social media tasks, including emotion recognition, hate speech detection, spam reviews detection, and hate speech spans detection. We conducted extensive analyses to demonstrate the efficiency of ViSoBERT on various Vietnamese social media characteristics, including emojis, teen-codes, and diacritics. Furthermore, our pre-trained language model ViSoBERT also shows the potential of leveraging Vietnamese social media text to enhance word representations compared to other PLMs. We hope the widespread use of our open-source ViSoBERT pre-trained language model will advance current NLP social media tasks and applications for Vietnamese. Other low-resource languages can adopt how to create PLMs for enhancing their current NLP social media tasks and relevant applications.

## Limitations

While we have demonstrated that ViSoBERT can perform state-of-the-art on a range of NLP social media tasks for Vietnamese, we think additional analyses and experiments are necessary to fully comprehend what aspects of ViSoBERT were responsible for its success and what understanding of Vietnamese social media texts ViSoBERT captures. We leave these additional investigations to future research. Moreover, future work aims to explore a broader range of Vietnamese social media downstream tasks that this paper may not cover. Also, we chose to train the base-size transformer model instead of the *Large* variant because base models are more accessible due to their lower computational requirements. For PhoBERT, XLM-R, and TwHIN-BERT, we implemented two versions *Base* and *Large* for all Vietnamese social media downstream tasks. However, it is not a fair comparison due to their significantly larger model configurations. Moreover, regular updates and expansions of the pre-training data are essential to keep up with the rapid evolution of social media. This allows the pre-trained model to adapt effectively to the dynamic linguistic patterns and trends in Vietnamese social media.

## Ethics Statement

The authors introduce ViSoBERT, a pre-trained language model for investigating social language phenomena in social media in Vietnamese. ViSoBERT is based on pre-training an existing pretrained language model (i.e., XLM-R), which lessens the influence of its construction on the environment. ViSoBERT makes use of a large-scale corpus of posts and comments from social communities that have been found to express harassment, bullying, incitement of violence, hate, offense, and abuse, as defined by the content policies of social media platforms, including Facebook, YouTube, and TikTok.

## Acknowledgement

This research was supported by The VNUHCM-University of Information Technology's Scientific Research Support Fund. We thank the anonymous EMNLP reviewers for their time and helpful suggestions that improved the quality of the paper.

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

## A  Tokenizations of the PLMs on Social Comments

We conducted an analysis of average token length by tasks of Pre-trained Language Models to provide insights into how different PLMs perform regarding token length across various Vietnamese social media downstream tasks. Figure 2 shows the average token length by Vietnamese social media downstream tasks of baseline PLMs and ours.

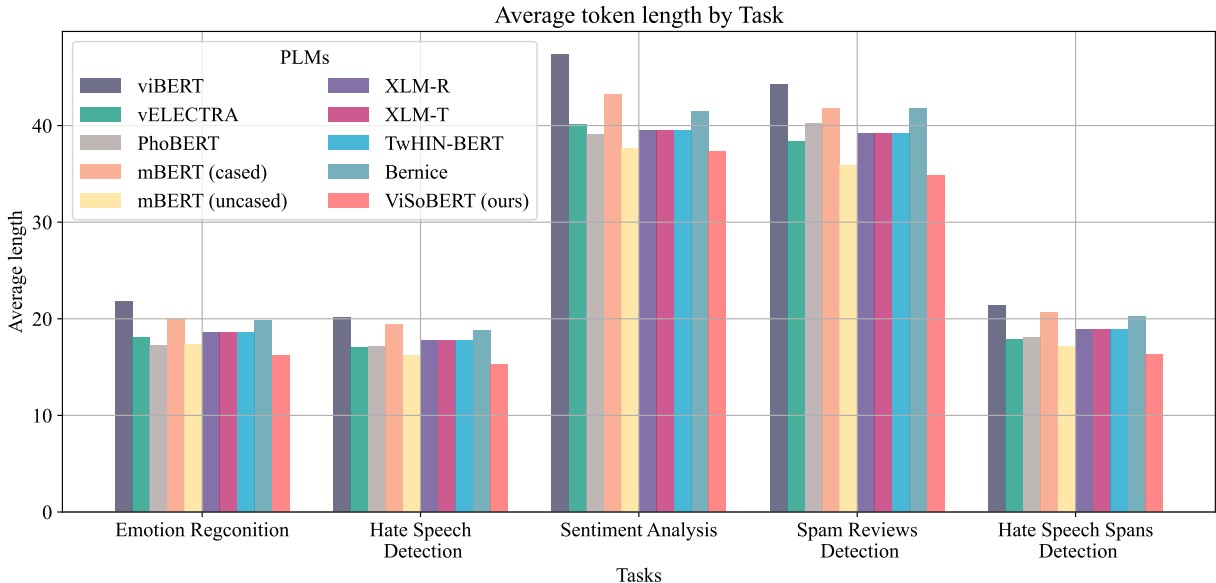

Figure 2: Average token length by tasks of PLMs.

## B  Experimental Settings

Following the hyperparameters in Table 6, we train our pre-trained language model ViSoBERT for Vietnamese social media texts.

|  | Algorithm | Adam |
|---|---|---|
|  | Learning rate | 5e-5 |
|  | Epsilon | 1e-8 |
| Optimizer | LR scheduler | linear decay and warmup |
|  | Warmup steps | 1000 |
|  | Betas | 0.9 and 0.99 |
|  | Weight decay | 0.01 |
|  | Sequence length | 128 |
| Batch | Batch size | 128 |
|  | Vocab size | 15002 |
| Misc | Dropout | 0.1 |
|  | Attention dropout | 0.1 |

Table 6: All hyperparameters established for training ViSoBERT.

## C PLMs with Pre-processing Techniques

For an in-depth understanding of the impact of social media texts on PLMs, we conducted an analysis of the test results on various processing aspects. Table 7 presents performances of the pre-trained language models on downstream Vietnamese social media tasks by applying word standardizing pre-processing techniques, while Table 8 presents performances of the pre-trained language models on downstream Vietnamese social media tasks by removing diacritics in all datasets.

| Model | Emotion Recognition | | | Hate Speech Detection | | | Sentiment Analysis | | | Spam Reviews Detection | | | Hate Speech Spans Detection | | |
|---|---|---|---|---|---|---|---|---|---|---|---|---|---|---|---|
| | Acc | WF1 | MF1 | Acc | WF1 | MF1 | Acc | WF1 | MF1 | Acc | WF1 | MF1 | Acc | WF1 | MF1 |
| PhoBERT$_{Large}$ | 64.94 | 64.85 | 62.71 | 87.68 | 87.25 | 65.41 | 76.80 | 76.61 | 76.61 | 89.47 | 89.41 | 76.12 | 91.73 | 91.62 | 86.59 |
| Δ | ↑ 0.23 | ↑ 0.19 | ↑ 0.16 | ↑ 0.36 | ↑ 0.27 | ↑ 0.27 | ↑ 0.28 | ↑ 0.25 | ↑ 0.25 | ↓ 0.65 | ↓ 0.62 | ↓ 0.76 | ↑ 0.29 | ↑ 0.16 | ↑ 0.03 |
| TwHIN-BERT$_{Large}$ | 64.42 | 64.46 | 61.28 | 87.82 | 87.28 | 65.68 | 77.17 | 76.94 | 76.94 | 89.49 | 89.43 | 76.35 | 91.74 | 91.64 | 86.67 |
| Δ | ↑ 0.21 | ↑ 0.17 | ↑ 0.16 | ↑ 0.59 | ↑ 0.50 | ↑ 0.45 | ↑ 0.25 | ↑ 0.11 | ↑ 0.11 | ↓ 0.98 | ↓ 0.99 | ↓ 0.93 | ↑ 0.29 | ↑ 0.17 | ↑ 0.02 |
| ViSoBERT [♣] | **68.25** | **68.52** | **65.94** | **88.53** | **88.33** | **68.82** | **78.01** | **77.88** | **77.88** | 90.83 | 90.75 | 78.77 | **91.89** | **91.82** | **86.93** |
| Δ | ↑ 0.15 | ↑ 0.15 | ↑ 0.06 | ↑ 0.02 | ↑ 0.02 | ↑ 0.08 | ↑ 0.18 | ↑ 0.13 | ↑ 0.13 | ↓ 0.16 | ↓ 0.17 | ↓ 0.29 | ↑ 0.27 | ↑ 0.25 | ↑ 0.13 |
| ViSoBERT [♦] | 68.10 | 68.37 | 65.88 | 88.51 | 88.31 | 68.74 | 77.83 | 77.75 | 77.75 | **90.99** | **90.92** | **79.06** | 91.62 | 91.57 | 86.80 |

Table 7: Performances of the pre-trained language models on downstream Vietnamese social media tasks by applying word standardizing pre-processing techniques. [♣] and [♦] denoted with and without standardizing word technique, respectively. Δ denoted the increase (↑) and the decrease (↓) in performances of the pre-trained language models compared to its competitors without normalizing teencodes.

To emphasize the essentials of diacritics, we conducted an analysis on several data samples by removing 100%, 75%, 50%, and 25% diacritics of total words that included diacritics in each comment of five downstream tasks. Table 8 presents performances of the pre-trained language models on downstream Vietnamese social media tasks by removing diacritics in all datasets.

| Model | Emotion Regconition | | | Hate Speech Detection | | | Sentiment Analysis | | | Spam Reviews Detection | | | Hate Speech Spans Detection | | |
|---|---|---|---|---|---|---|---|---|---|---|---|---|---|---|---|
| | Acc | WF1 | MF1 | Acc | WF1 | MF1 | Acc | WF1 | MF1 | Acc | WF1 | MF1 | Acc | WF1 | MF1 |
| *Removing 100% diacritics in each comment* | | | | | | | | | | | | | | | |
| PhoBERT$_{Large}$ | 49.35 | 49.18 | 43.95 | 81.25 | 81.42 | 55.43 | 62.38 | 62.36 | 62.36 | 87.68 | 87.56 | 71.89 | 91.32 | 91.37 | 86.43 |
| Δ | ↓15.36 | ↓15.48 | ↓18.60 | ↓ 6.07 | ↓ 5.56 | ↓ 9.71 | ↓14.14 | ↓14.00 | ↓13.86 | ↓ 2.44 | ↓ 2.47 | ↓ 4.99 | ↓ 0.12 | ↓ 0.09 | ↓ 0.13 |
| TwHIN-BERT$_{Large}$ | 49.32 | 49.15 | 43.52 | 84.25 | 78.48 | 51.32 | 66.66 | 66.68 | 66.68 | 89.45 | 89.26 | 74.59 | 91.12 | 91.22 | 86.33 |
| Δ | ↓14.89 | ↓15.14 | ↓17.60 | ↓ 2.98 | ↓ 8.30 | ↓12.99 | ↓10.26 | ↓10.15 | ↓10.15 | ↓ 1.02 | ↓ 1.16 | ↓ 2.69 | ↓ 0.33 | ↓ 0.25 | ↓ 0.32 |
| ViSoBERT [♣] | 61.96 | 62.05 | 58.48 | 87.29 | 86.76 | 64.87 | 72.95 | 72.91 | 72.91 | 89.75 | 89.72 | 76.12 | 91.48 | 91.42 | 86.69 |
| Δ | ↓ 6.14 | ↓ 6.32 | ↓ 7.40 | ↓ 1.22 | ↓ 1.55 | ↓ 3.90 | ↓ 4.88 | ↓ 4.84 | ↓ 4.84 | ↓ 1.24 | ↓ 1.20 | ↓ 2.94 | ↓ 0.14 | ↓ 0.15 | ↓ 0.11 |
| *Removing 75% diacritics in each comment* | | | | | | | | | | | | | | | |
| PhoBERT$_{Large}$ | 51.94 | 51.79 | 47.79 | 84.74 | 84.03 | 58.37 | 66.00 | 65.98 | 65.98 | 88.23 | 88.12 | 72.42 | 90.38 | 90.23 | 85.42 |
| Δ | ↓12.77 | ↓12.87 | ↓14.76 | ↓ 2.58 | ↓ 2.95 | ↓ 6.77 | ↓10.52 | ↓10.38 | ↓10.24 | ↓ 1.89 | ↓ 1.91 | ↓ 4.46 | ↓ 1.06 | ↓ 1.23 | ↓ 1.14 |
| TwHIN-BERT$_{Large}$ | 51.32 | 51.17 | 44.63 | 83.22 | 81.42 | 52.24 | 67.23 | 67.32 | 67.32 | 89.12 | 88.95 | 75.20 | 90.62 | 89.93 | 85.81 |
| Δ | ↓12.89 | ↓13.12 | ↓16.49 | ↓ 4.01 | ↓ 5.36 | ↓12.99 | ↓ 9.69 | ↓ 9.51 | ↓ 9.51 | ↓ 1.35 | ↓ 1.47 | ↓ 2.08 | ↓ 0.83 | ↓ 1.54 | ↓ 0.84 |
| ViSoBERT [♠] | 62.34 | 62.26 | 58.13 | 87.35 | 86.88 | 65.12 | 73.90 | 73.97 | 73.97 | 90.41 | 90.31 | 76.17 | 91.02 | 91.17 | 86.02 |
| Δ | ↓ 5.76 | ↓ 6.11 | ↓ 7.75 | ↓ 1.16 | ↓ 1.43 | ↓ 3.65 | ↓ 3.93 | ↓ 3.78 | ↓ 3.78 | ↓ 0.58 | ↓ 0.61 | ↓ 2.89 | ↓ 0.60 | ↓ 0.40 | ↓ 0.78 |
| *Removing 50% diacritics in each comment* | | | | | | | | | | | | | | | |
| PhoBERT$_{Large}$ | 57.28 | 57.36 | 54.02 | 85.29 | 84.71 | 59.40 | 66.57 | 66.46 | 66.46 | 89.02 | 88.81 | 73.10 | 90.42 | 90.47 | 85.62 |
| Δ | ↓ 7.43 | ↓ 7.30 | ↓ 8.53 | ↓ 2.03 | ↓ 2.27 | ↓ 5.74 | ↓ 9.95 | ↓ 9.90 | ↓ 9.76 | ↓ 1.10 | ↓ 1.22 | ↓ 3.78 | ↓ 1.02 | ↓ 0.99 | ↓ 0.94 |
| TwHIN-BERT$_{Large}$ | 53.70 | 53.39 | 49.55 | 83.41 | 83.31 | 55.22 | 70.42 | 70.53 | 70.53 | 89.33 | 89.05 | 75.32 | 90.73 | 90.12 | 85.92 |
| Δ | ↓10.51 | ↓10.90 | ↓11.57 | ↓ 3.82 | ↓ 3.47 | ↓10.61 | ↓ 6.30 | ↓ 6.30 | ↓ 6.30 | ↓ 1.14 | ↓ 1.37 | ↓ 1.96 | ↓ 0.72 | ↓ 1.35 | ↓ 0.73 |
| ViSoBERT [♥] | 62.96 | 62.87 | 60.55 | 87.44 | 87.10 | 65.25 | 74.76 | 74.72 | 74.72 | 90.41 | 90.35 | 77.31 | 91.12 | 91.24 | 86.22 |
| Δ | ↓ 5.14 | ↓ 5.50 | ↓ 5.33 | ↓ 1.07 | ↓ 1.21 | ↓ 3.52 | ↓ 3.07 | ↓ 3.03 | ↓ 3.03 | ↓ 0.58 | ↓ 0.57 | ↓ 1.75 | ↓ 0.50 | ↓ 0.33 | ↓ 0.58 |
| *Removing 25% diacritics in each comment* | | | | | | | | | | | | | | | |
| PhoBERT$_{Large}$ | 61.03 | 60.80 | 57.87 | 85.97 | 85.51 | 61.96 | 73.42 | 73.28 | 73.28 | 89.80 | 89.59 | 75.53 | 90.63 | 90.69 | 85.76 |
| Δ | ↓ 3.68 | ↓ 3.86 | ↓ 4.68 | ↓ 1.35 | ↓ 1.47 | ↓ 3.18 | ↓ 3.10 | ↓ 3.08 | ↓ 2.94 | ↓ 0.32 | ↓ 0.44 | ↓ 1.35 | ↓ 0.81 | ↓ 0.77 | ↓ 0.80 |
| TwHIN-BERT$_{Large}$ | 61.18 | 60.98 | 57.42 | 86.85 | 86.13 | 63.14 | 73.21 | 73.11 | 73.11 | 89.91 | 89.43 | 76.32 | 91.09 | 90.72 | 86.02 |
| Δ | ↓ 3.03 | ↓ 3.31 | ↓ 3.70 | ↓ 0.38 | ↓ 0.65 | ↓ 2.09 | ↓ 3.71 | ↓ 3.72 | ↓ 3.72 | ↓ 0.56 | ↓ 0.99 | ↓ 0.96 | ↓ 0.36 | ↓ 0.75 | ↓ 0.63 |
| ViSoBERT [✣] | 64.64 | 64.53 | 61.29 | 87.85 | 87.56 | 66.54 | 75.42 | 75.44 | 75.44 | 90.76 | 90.64 | 78.15 | 91.22 | 91.24 | 86.47 |
| Δ | ↓ 3.43 | ↓ 3.84 | ↓ 4.59 | ↓ 0.66 | ↓ 0.75 | ↓ 2.23 | ↓ 2.41 | ↓ 2.31 | ↓ 2.31 | ↓ 0.23 | ↓ 0.28 | ↓ 0.91 | ↓ 0.40 | ↓ 0.33 | ↓ 0.33 |
| ViSoBERT [♦] | **68.10** | **68.37** | **65.88** | **88.51** | **88.31** | **68.77** | **77.83** | **77.75** | **77.75** | **90.99** | **90.92** | **79.06** | **91.62** | **91.57** | **86.80** |

Table 8: Performances of the pre-trained language models on downstream Vietnamese social media tasks by removing diacritics in all datasets. [♣], [♠], [♥], [✣] and [♦] denoted the performances of our pre-trained on removing 100%, 75%, 50%, 25% in each comment, respectively, and not removing diacritics marks dataset, respectively. Δ denoted the increase (↑) and the decrease (↓) in performances of the pre-trained language models compared to its competitors without removing diacritics marks.

# D  PLM-based Features for BiLSTM and BiGRU

We conduct experiments with various models, including BiLSTM and BiGRU, to understand better the word embedding feature extracted from the pre-trained language models. Table 9 shows performances of the pre-trained language model as input features to BiLSTM and BiGRU on downstream Vietnamese social media tasks.

| Model | Emotion Regconition | | | Hate Speech Detection | | | Sentiment Analysis | | | Spam Reviews Detection | | | Hate Speech Spans Detection | | |
|---|---|---|---|---|---|---|---|---|---|---|---|---|---|---|---|
| | Acc | WF1 | MF1 | Acc | WF1 | MF1 | Acc | WF1 | MF1 | Acc | WF1 | MF1 | Acc | WF1 | MF1 |
| *BiLSTM* | | | | | | | | | | | | | | | |
| PhoBERT$_{Large}$ | 57.58 | 56.65 | 50.55 | 86.11 | 84.04 | 56.03 | 69.71 | 69.70 | 69.70 | 87.80 | 87.10 | 68.95 | 84.01 | 80.70 | 74.35 |
| TwHIN-BERT$_{Large}$ | 61.47 | 61.31 | 56.73 | 83.14 | 82.72 | 55.84 | 64.76 | 64.82 | 64.82 | 88.73 | 88.23 | 72.18 | 85.92 | 84.43 | 78.28 |
| ViSoBERT | **63.06** | **62.36** | **59.16** | **87.62** | **86.81** | **64.82** | **73.52** | **73.50** | **73.50** | **90.11** | **89.79** | **75.71** | **88.37** | **87.87** | **82.18** |
| *BiGRU* | | | | | | | | | | | | | | | |
| PhoBERT$_{Large}$ | 55.12 | 54.53 | 49.59 | 85.21 | 83.23 | 54.59 | 70.01 | 70.01 | 70.01 | 86.06 | 84.89 | 62.54 | 84.23 | 81.01 | 74.57 |
| TwHIN-BERT$_{Large}$ | 60.46 | 60.30 | 55.23 | 85.73 | 83.45 | 54.74 | 63.11 | 61.39 | 61.39 | 87.67 | 86.38 | 66.83 | 86.10 | 84.52 | 78.49 |
| ViSoBERT | **63.20** | **63.25** | **60.73** | **87.02** | **86.25** | **63.36** | **70.48** | **70.53** | **70.53** | **89.33** | **88.98** | **76.57** | **88.88** | **88.19** | **82.63** |

Table 9: Performances of the pre-trained language models as input features to BiLSTM and BiGRU on downstream Vietnamese social media tasks.

We implemented various PLMs when used as input features in combination with BiLSTM and BiGRU models to verify the ability to extract Vietnamese social media texts. The evaluation is conducted on the dev set, and the performance is measured per epoch for downstream tasks. Table 9 shows performances of the PLMs as input features to BiLSTM and BiGRU on the dev set per epoch.

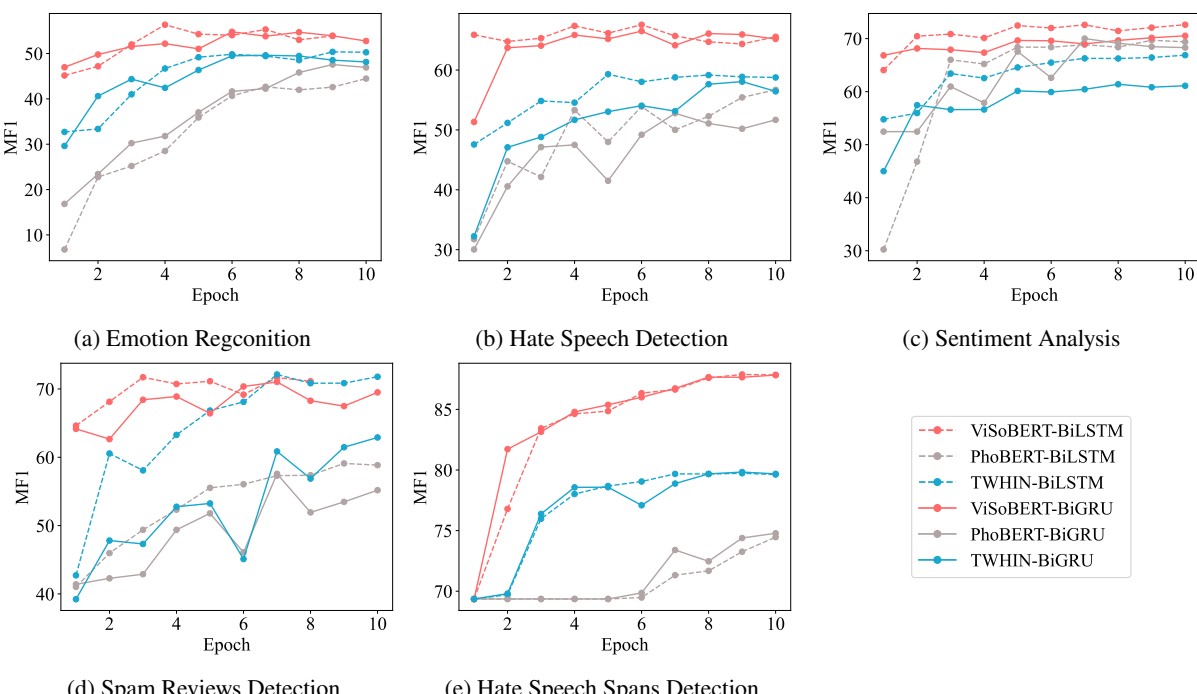

(a) Emotion Regconition     (b) Hate Speech Detection     (c) Sentiment Analysis

(d) Spam Reviews Detection     (e) Hate Speech Spans Detection

Figure 3: Performances of the PLMs as input features to BiLSTM and BiGRU on the dev set per epoch on Vietnamese social media downstream tasks. *Large* versions of PhoBERT and TwHIN-BERT are implemented for these experiments.

# E  Updating New Spans of Hate Speech Span Detection Samples with Pre-processing Techniques

Due to performing pre-processing techniques, the span positions on the data samples can be changed. Therefore, we present Algorithm 1, which shows how to update new span positions of samples applied with pre-processing techniques in the Hate Speech Spans Detection task (UIT-ViHOS dataset). This algorithm takes as input a comment and its spans and returns the pre-processed comment and its span along with pre-processing techniques.

---

**Algorithm 1** Updating new spans of samples applied with pre-processing techniques in Hate Speech Spans Detection task (UIT-ViHOS dataset).

---

1: **procedure** ALGORITHM(comment, label, delete)
2:  **assert** len(comment) == len(label)
3:  new_comment ← [ ], new_label ← [ ]
4:  **for** $i$ ← 0 **to** len(comment) **do**
5:    check ← 0
6:    **if** comment[$i$] **in** emoji_to_word.keys() **then**
7:      **if** delete **then**
8:        **continue**
9:      **for** $j$ ← 0 **to** len(emoji_to_word[comment[i]].split(' ')) **do**
10:        **if** label[$i$] == 'B-T' **then**
11:          **if** check == 0 **then**
12:            check ← check + 1, new_label.append(label[i])
13:          **else**
14:            new_label.append('I-T')
15:        **else**
16:          new_label.append(label[i])
17:        new_comment.append(emoji_to_word[comment[i]].split(' ')[j])
18:    **else**
19:      new_sentence.append(comment[i])
20:      new_label.append(label[i])
21:  **assert** len(new_comment) == len(new_label)
22:  **return** new_comment, new_label

---

# F  Tokenizations of the PLMs on Removing Diacritics Social Comments

We analyze several data samples to see the tokenized ability of Vietnamese social media textual data while removing diacritics on comments. Table 10 shows several non-diacritics Vietnamese social comments and their tokenizations with the tokenizers of the three best pre-trained language models, ViSoBERT (ours), PhoBERT, and TwHIN-BERT.

| Model | Example 1 | Example 2 |
|---|---|---|
| Raw comment | cái con đồ chơi đó mua ở đâu nhỉ . cười đéo nhặt được mồm 😂😂😂

*English*: where did you buy that toy . LMAO 😂😂😂 | Ôi bố cái lũ thanh niên hãm lol. Đẹp mặt quá 😒😒

*English*: Oh my god damn teenagers, lol. So deserved 😒😒 |
| **_Removing 100% diacritics in each comment_** | | |
| Comment | cai con do choi do mua o dau nhi . cuoi deo nhat duoc mom . 😂😂😂 | Oi bo cai lu thanh nien ham lol. Dep mat qua 😒😒 |
| PhoBERT | , "c a i", "c o n", "d o", "c h o @ @", "i", "d o", "m u a", "o", "d @ @", "a u", "n h i", ".", "c u @ @", "o i", "d @ @", "e o", "n h @ @", "a t", "d u @ @", "o c", "m o m", ".", <unk>, <unk>, <unk>,  | , "O @ @", "i", "b o", "c a i", "l u", "t h a n h", "n i @ @", "e n", "h a m", "l o @ @", "l", ".", "D e @ @", "p", "m a t", "q u a", <unk>, <unk>,  |
| TwHIN-BERT | , "cai", "con", "do", "cho", "i", "do", "mua", "o", "dau", "nhi", "", ".", "cu", "oi", "de", "o", "nha", "t", "du", "oc", "mom", "", ".", "", "😂", "😂", "😂",  | , "Oi", "bo", "cai", "lu", "thanh", "nie", "n", "ham", "lol", ".", "De", "p", "mat", "qua", "", "😒", "😒",  |
| ViSoBERT | , "cai", "con", "do", "choi", "do", "mua", "o", "dau", "nhi", ".","cu", "oi", "d", "eo", "nhat", "duoc", "m", "om", ".", "😂😂😂",  | , "O", "i", "bo", "cai", "lu", "thanh", "ni", "en", "h", "am", "lol", ".", "D", "ep", "mat", "qua", "😒😒",  |
| **_Removing 75% diacritics in each comment_** | | |
| Comment | cai con do chơi do mua o đâu nhi . cười deo nhat duoc mom . 😂😂😂 | Ôi bo cai lu thanh niên hãm lol. Dep mat qua 😒😒 |
| PhoBERT | , "c a i", "c o n", "d o", "c h ơ i", "d o", "m u a", "o", "đ â u", "n h i", ".", "c ư ờ i", "d @ @", "e o", "n h @ @", "a t", "d u @ @", "o c", "m o m", ".",<unk>, <unk>, <unk>,  | , "Ô i", "b o", "c a i", "l u", "t h a n h _ n i ê n", "h ã m", "l o @ @", "l", ".", "D e @ @", "p", "m a t", "q u a", <unk>, <unk>,  |
| TwHIN-BERT | , "cai", "con", "do", "chơi", "do", "mua", "o", "đâu", "nhi", "", ".", "cười", "de", "o", "nha", "t", "du", "oc", "mom", "", ".", "", "😂", "😂", "😂",  | , "Ô", "i", "bo", "cai", "lu", "thanh", "niên", "", "hã", "m", "lol", ".", "De", "p", "mat", "qua", "", "😒", "😒",  |
| ViSoBERT | , "cai", "con", "do", "choi", "do", "mua", "o", "đâu", "nhi", ".", "cười", "d", "eo", "nhat", "duoc", "m", "om", ".", "😂😂😂",  | , "Ôi", "bo", "cai", "lu", "thanh", "n", "iên", "hã", "m", "lol", ".", "D", "ep", "mat", "qua", "😒😒",  |
| **_Removing 50% diacritics in each comment_** | | |
| Comment | cai con do chơi do mua o đâu nhỉ . cười đéo nhặt duoc mom . 😂😂😂 | Ôi bo cai lu thanh niên hãm lol. Dep mặt quá 😒😒 |
| PhoBERT | , "c a i", "c o n", "d o", "c h ơ i", "d o", "m u a", "o", "đ â u", "n h ỉ", ".", "c ư ờ i", "đ @ @", "é o", "n h ặ t", "d u @ @", "o c", "m o m", ".", <unk>, <unk>, <unk>,  | , "Ô i", "b o", "c a i", "l u", "t h a n h _ n i ê n", "h ã m", "l o @ @", "l", ".", "D e @ @", "p", "m ặ t", "q u á", <unk>, <unk>,  |
| TwHIN-BERT | , "cai", "con", "do", "chơi", "do", "mua", "o", "đâu", "nhỉ", "", ".", "cười", "đ", "é", "o", "nh", "ặt", "du", "oc", "mom", "", ".", "", "😂", "😂", "😂",  | , "Ô", "i", "bo", "cai", "lu", "thanh", "niên", "", "hã", "m", "lol", ".", "De", "p", "mặt", "quá", "", "😒", "😒",  |
| ViSoBERT | , "cai", "con", "do", "chơi", "do", "mua", "o", "đâu", "nhỉ", ".", "cười", "đéo", "nh", "ặt", "duoc", "m", "om", ".", "😂😂😂",  | , "Ôi", "bo", "cai", "lu", "thanh", "n", "iên", "hã", "m", "lol", ".", "D", "ep", "mặt", "quá", "😒😒",  |
| **_Removing 25% diacritics in each comment_** | | |
| Comment | cai con do chơi đó mua ở đâu nhỉ . cười đéo nhặt duoc mồm . 😂😂😂 | Ôi bo cai lu thanh niên hãm lol. Đep mặt quá 😒😒 |
| PhoBERT | , "c a i", "c o n", "d o", "c h ơ i", "đ ó", "m u a", "ở", "đ â u", "n h ỉ", ".", "c ư ờ i", "đ @ @", "é o", "n h ặ t", "d u @ @", "o c", "m ồ m", ".", <unk>, <unk>, <unk>,  | , "Ô i", "b o", "c a i", "l u", "t h a n h _ n i ê n", "h ã m", "l o @ @", "l", ".", "Đ e p _ @ @", "m ặ t", "q u á", <unk>, <unk>,  |
| TwHIN-BERT | , "cai", "con", "do", "chơi", "do", "mua", "o", "đâu", "nhỉ", "", ".", "cười", "đ", "é", "o", "nh", "ặt", "du", "oc", "mom", "", ".", "", "😂", "😂", "😂",  | , "Ô", "i", "bo", "cai", "lu", "thanh", "niên", "", "hã", "m", "lol", ".", "Đep", "mặt", "quá", "", "😒", "😒",  |
| ViSoBERT | , "cai", "con", "do", "chơi", "đó", "mua", "ở", "đâu", "nhỉ", ".", "cười", "đéo", "nh", "ặt", "duoc", "mồm", ".", "😂😂😂",  | , "Ôi", "bo", "cai", "lu", "thanh", "n", "iên", "hã", "m", "lol", ".", "Đep", "mặt", "quá", "😒😒",  |

Table 10: Actual social comments and their tokenizations with the tokenizers of the three pre-trained language models, including PhoBERT, TwHIN-BERT, and ViSoBERT, on removing diacritics of social comments. **Disclaimer**: This table contains actual comments on social networks that might be construed as abusive, offensive, or obscene.