# OpenReview forum: "ViSoBERT: A Pre-Trained Language Model for Vietnamese Social Media Text Processing"
_EMNLP/2023/Conference — EMNLP 2023 Main_

### Official Review · Reviewer_117r · 2023-08-02

**Soundness:** 4

**Excitement:**

5: Transformative: This paper is likely to change its subfield or computational linguistics broadly. It should be considered for a best paper award. This paper changes the current understanding of some phenomenon, shows a widely held practice to be erroneous in someway, enables a promising direction of research for a (broad or narrow) topic, or creates an exciting new technique.

**Paper Topic And Main Contributions:**

In this paper, the authors introduce PLM4ViSMT, the initial monolingual pre-trained language model designed specifically for Vietnamese social media texts. This model, based on the XLM-R architecture, is trained on a vast and varied collection of high-quality Vietnamese social media texts. They tested the pre-existing model on five significant natural language tasks related to Vietnamese social media content. These tasks included identifying emotions, detecting hate speech, analyzing sentiment, detecting spam reviews, and identifying hate speech segments.


**Reasons To Accept:**

They introduce the initial PLM that utilizes the XLM-R architecture and pretraining method specifically designed for handling Vietnamese social media text.

**Reasons To Reject:**

-

**Reproducibility:**

4: Could mostly reproduce the results, but there may be some variation because of sample variance or minor variations in their interpretation of the protocol or method.

**Reviewer Confidence:**

4: Quite sure. I tried to check the important points carefully. It's unlikely, though conceivable, that I missed something that should affect my ratings.

---

> ### Author Rebuttal · Authors · 2023-08-27
>
> Dear Reviewer ${\color{Green}117r}$,
>
> We would like to thank reviewer ${\color{Green}117r}$ for your really positive reviews. We firmly believe that conducting additional experiments and adding supplementary information based on the comments and suggestions from reviewers ${\color{Orange}mcGf}$ and ${\color{Blue}iNVu}$ would significantly enhance the quality of our paper. We're committed to making these improvements to create a better version of our work.

---

### Official Review · Reviewer_mcGf · 2023-08-06

**Soundness:** 3

**Excitement:**

3: Ambivalent: It has merits (e.g., it reports state-of-the-art results, the idea is nice), but there are key weaknesses (e.g., it describes incremental work), and it can significantly benefit from another round of revision. However, I won't object to accepting it if my co-reviewers champion it.

**Paper Topic And Main Contributions:**

Authors present a new pretrained LM, PLM4ViSMT, trained on Vietnamese social media text extracted from social media networks - Facebook, TikTok and YouTube. Authors compare previous pre-trained PLMs performance on downstream task with that of PLM4ViSMT.

**Questions For The Authors:**

- Sec 3.1 : How were Vietnamese social media text identified?
- Sec 4.2 : Were the results computed based on single runs or average of multple runs?
- How did training loss compare with continued pre-training of other Vietnamese Models? Instead of training from sratch continued pretraining of previous models would achieve better results?
- Further the baseline models when continued pre-trained usiing the same data can still perform better , with lesser training data.

**Reasons To Accept:**

- Authors propose a new model suited for Vietnamese social media text processing needs. Model presented outperforms other models specific to Vietnamese language.
- Authors conduct extensive experiments on downstream tasks to showcase improvements on downstream tasks.

**Reasons To Reject:**

- Insufficient details on how Vietnamese social media text were identified and collected.
- Details regarding the behavior during training would be useful in assessing the relative improvements in the model due to training.
- Baselines : Comparing other multilingual pre-trained models trained on social media text would be better benchmarks -  like BERNICE.


**Reproducibility:**

2: Would be hard pressed to reproduce the results. The contribution depends on data that are simply not available outside the author's institution or consortium; not enough details are provided.

**Reviewer Confidence:**

3: Pretty sure, but there's a chance I missed something. Although I have a good feel for this area in general, I did not carefully check the paper's details, e.g., the math, experimental design, or novelty.

**Typos Grammar Style And Presentation Improvements:**

- L215: 01 GB of uncompressed texts? I am assuming this is a typo.
- Including some more details on the training performance and plots would help in following up on the research presented in the paper.

---

> ### Author Rebuttal · Authors · 2023-08-27
>
> Dear Reviewer ${\color{Orange}mcGf}$,
>
> We thank the reviewer ${\color{Orange}mcGf}$ for taking the time to read this paper and suggest several valuable comments. Please find our detailed responses below.
>
> **Q: Insufficient details on how Vietnamese social media text were identified and collected.**
>
> As mentioned in the paper, our pretraining data is collected from the three most popular Vietnamese social media platforms, including Facebook, TikTok, and YouTube. To effectively gather data from these platforms, we harnessed the capabilities of specialized tools provided by each platform:
>
> 1. Facebook: We crawled comments from Vietnamese verified pages by Facebook (including government pages, celebrity pages, organization pages, etc.) posts via the *Facebook Graph API* between January 2016 and December 2022.
> 2. Tiktok: We collected comments from Vietnamese-verified channels via *Tiktok Research API* between January 2020 and December 2022.
> 3. Youtube: We scrapped comments from Vietnam-verified channels' videos via *Youtube Data API* between January 2016 and December 2022.
>
> As a result, our pretraining data after crawling and preprocessing contains 1GB of uncompressed text. Our pretraining data is available only for research purposes.
>
> Note that a verified page means the platform has confirmed that the Page or profile is the authentic presence of the individual, public figure, or brand it represents and verified channels.
>
> **Q: Details regarding the behavior during training would be useful in assessing the relative improvements in the model due to training.**
>
> Thanks a lot for your helpful suggestion! We're excited about trying out these extra experiments in our future work. Your insightful suggestion has given us a great new direction to explore.
>
> **Q: Baselines : Comparing other multilingual pre-trained models trained on social media text would be better benchmarks - like BERNICE.**
>
> In the paper, we implemented TwHIN-BERT, a multilingual PLM for social media tasks (including Vietnamese), and other SOTA pre-trained language models for Vietnamese such as PhoBERT, XLM-R, etc. However, we agree with the reviewer that more LMs, specifically for Social Media tasks, must be implemented and compared to our PLM4ViSMT. We conducted additional experiments regarding your concerns by comparing our pre-trained model with XLM-T [1] and Bernice [2] (multilingual PLMs for social media, including Vietnamese).
>
> Our results indicate that our PLM4ViSMT continues to exhibit superior performance compared to both XLM-T and Bernice across 5 Vietnamese social media tasks. XLM-T, TwHIN-BERT, and Bernice, like TwHIN-BERT, were exclusively trained on pretraining data from the Twitter platform. However, this limitation hinders these models from effectively capturing the intricate nuances of the Vietnamese social media context. The primary reason is that Twitter isn't a widely utilized social media platform in Vietnam, and therefore, the training data from this source may not accurately encapsulate the linguistic intricacies and contextual subtleties prevalent within the Vietnamese social media landscape. See the table below for the full results.
>
> | **Model**      | **Avg** | **Emotion Recognition** |        |        | **Hate Speech Detection** |        |        | **Sentiment Analysis** |        |        | **Spam Reviews Detection** |        |        | **Hate Speech Spans Detection** |        |        |
> | :------------- | :-----: | :---------------------: | :----: | :----: | :-----------------------: | :----: | :----: | :--------------------: | :----: | :----: | :------------------------: | :----: | :----: | :-----------------------------: | :----: | :----: |
> |                |         | Acc                     | WF1    | MF1    | Acc                       | WF1    | MF1    | Acc                    | WF1    | MF1    | Acc                        | WF1    | MF1    | Acc                             | WF1    | MF1    |
> | XLM-T          | 72\.23  | 64\.64                  | 64\.37 | 59\.86 | 86\.22                    | 86\.12 | 63\.48 | 75\.66                 | 75\.6  | 75\.6  | 90\.07                     | 90\.11 | 76\.66 | 90\.88                          | 90\.88 | 85\.53 |
> | Bernice        | 72\.49  | 64\.21                  | 64\.27 | 60\.68 | 86\.12                    | 86\.48 | 64\.32 | 74\.57                 | 74\.90 | 74\.90 | 90\.22                     | 90\.21 | 76\.89 | 90\.48                          | 90\.06 | 85\.67 |
> | **PLM4ViSMT** | **75\.65**  | 68\.10                  | 68\.37 | 65\.88 | 88\.51                    | 88\.31 | 68\.77 | 77\.83                 | 77\.75 | 77\.75 | 90\.99                     | 90\.92 | 79\.06 | 91\.62                          | 91\.57 | 86\.80 |
>
> Since Bernice is the only multilingual model trained from scratch on Twitter data with a custom tokenizer, we also conducted experiments to verify its tokenizer on Vietnamese social media text. Unsurprisingly, our experiments revealed that Bernice's tokenizer performs poorly compared to most of our established baseline models, excluding viBERT and mBERT. This outcome further emphasizes the significance of our PLM4ViSMT's specialized tokenizer in effectively handling Vietnamese social media text (which is showcased in Section 3.3, "The Vietnamese Social Media Tokenizer" of the paper). See the table below for the results.
>
> | **Model**      | **Avg** | **Emotion Recognition** | **Hate Speech Detection** | **Sentiment Analysis** | **Spam Reviews Detection** | Hate Speech Spans Detection |
> | :------------- | :-----: | :---------------------: | :-----------------------: | :--------------------: | :------------------------: | :------------------------------: |
> | XLM-T          | 26\.77  | 18\.59                  | 17\.74                    | 39\.45                 | 39\.21                     | 18\.87                           |
> | Bernice        | 28\.41  | 19\.87                  | 18\.82                    | 41\.41                 | 41\.74                     | 20\.20                           |
> | **PLM4ViSMT** | **23\.98**  | 16\.23                  | 15\.30                    | 37\.30                 | 34\.81                     | 16\.26                           |
>
> Upon publication, we will move these results forward to the main experimental results using the extra page of the camera-ready version.
>
> **References**
>
> [1] Barbieri et al., "XLM-T: Multilingual Language Models in Twitter for Sentiment Analysis and Beyond" LREC 2022
>
> [2] DeLucia et al., "Bernice: A Multilingual Pre-trained Encoder for Twitter" EMNLP 2022
>
> **Q: Sec 4.2 : Were the results computed based on single runs or average of multple runs?**
>
> All of our results were computed based on a single run using the same seed for consistency across all our experiments.  However, it's important to note that we also undertook an extensive series of ablation studies to further validate the exceptional performance of our pre-trained model in comparison to other approaches. These ablation studies were meticulously designed to isolate and analyze the impact of different factors.
>
> By carrying out these ablation studies, we could showcase not only the outstanding performance of our pre-trained model but also provide insights into its inner workings. The results of these studies collectively support our claim of superiority over other existing models and underscore the comprehensive nature of our pre-trained model.
>
> **Q: L215: 01 GB of uncompressed texts? I am assuming this is a typo.**
>
> We thank you for pointing this typo out. We noted that our pretraining data contains only 1GB of uncompressed text. We have revised it.
>
> We are grateful to the reviewers for your valuable comments and recommendations. These comments helped us to improve the quality of the article better. We hope the reviewers will be satisfied with our revisions according to your comments and recommendations.

---

### Official Review · Reviewer_iNVu · 2023-08-11

**Soundness:** 3
**Typos Grammar Style And Presentation Improvements:** Line 215

**Excitement:**

4: Strong: This paper deepens the understanding of some phenomenon or lowers the barriers to an existing research direction.

**Paper Topic And Main Contributions:**

This paper proposes a PLM for Vietnamese Social Media. It is the first model based on the XLM-R architecture. They achieve SOTA results across different social media downstream tasks. Their model is the first one with custom tokeniser for Vietnamese social media. They have reported an extensive study to analyse their model which include different variations of masking, including emojis, teencode, and diacritics etc.

**Reasons To Accept:**

1. This paper focuses on building PLM for social media task in Vietnamese. Although the language is spoken by 100M people, not a lot of progress has been done in the field of social media tasks.
2. This is the first monolingual PLM focusing on social media task in Vietnamese and have achieved SOTA results for Emotion Recognition, Hate Speech Detection, Spam Reviews Detection, Hate Speech Spans Detection. Although it doesn't achieve SOTA results for Sentiment Analysis task, it outperforms the existing general Vietnamese LM.
3. Detailed ablation studies is carried out to show importance of different aspects of model like including emojis, removing emojis, converting emojis to text.

**Reasons To Reject:**

1. There is less information regarding the data scrapped for pretraining.
2. The paper didn't specify if the data is going to be open sourced.
3. The PLM proposed in the paper should have been compared with other LM trained specifically for Social Media tasks.

**Reproducibility:**

2: Would be hard pressed to reproduce the results. The contribution depends on data that are simply not available outside the author's institution or consortium; not enough details are provided.

**Reviewer Confidence:**

3: Pretty sure, but there's a chance I missed something. Although I have a good feel for this area in general, I did not carefully check the paper's details, e.g., the math, experimental design, or novelty.

---

> ### Author Rebuttal · Authors · 2023-08-27
>
> Dear Reviewer ${\color{Blue}iNVu}$,
>
> We thank the reviewer ${\color{Blue}iNVu}$ for taking the time to read this paper and suggest several valuable comments. Please find our detailed responses below. Please find our detailed response below.
>
> **Q: There is less information regarding the data scrapped for pretraining.**
>
> As mentioned in the paper, our pretraining data is collected from the three most popular Vietnamese social media platforms, including Facebook, TikTok, and YouTube. To effectively gather data from these platforms, we harnessed the capabilities of specialized tools provided by each platform:
>
> 1. Facebook: We crawled comments from Vietnamese verified pages by Facebook (including government pages, celebrity pages, organization pages, etc.) posts via the *Facebook Graph API* between January 2016 and December 2022.
> 2. Tiktok: We collected comments from Vietnamese-verified channels via *Tiktok Research API* between January 2020 and December 2022.
> 3. Youtube: We scrapped comments from Vietnam-verified channels' videos via *Youtube Data API* between January 2016 and December 2022.
>
> As a result, our pretraining data after crawling and preprocessing contains 1GB of uncompressed text. Our pretraining data is available only for research purposes.
>
> Note that a verified page means the platform has confirmed that the Page or profile is the authentic presence of the individual, public figure, or brand it represents and verified channels.
>
> **Q: The paper didn't specify if the data is going to be open sourced.**
>
> Thank you for your insightful suggestion. In the paper, lines 27-28, we noted that "Our PLM4ViSMT model is available only for research purposes." we noted that pretraining data would also be open-sourced.
>
> **Q: The PLM proposed in the paper should have been compared with other LM trained specifically for Social Media tasks.**
>
> In the paper, we implemented TwHIN-BERT, a multilingual PLM for social media tasks (including Vietnamese), and other SOTA pre-trained language models for Vietnamese such as PhoBERT, XLM-R, etc. However, we agree with the reviewer that more LMs, specifically for Social Media tasks, must be implemented and compared to our PLM4ViSMT. We conducted additional experiments regarding your concerns by comparing our pre-trained model with XLM-T [1] and Bernice [2] (multilingual PLMs for social media, including Vietnamese).
>
> Our results indicate that our PLM4ViSMT continues to exhibit superior performance compared to both XLM-T and Bernice across 5 Vietnamese social media tasks. XLM-T, TwHIN-BERT, and Bernice, like TwHIN-BERT, were exclusively trained on pretraining data from the Twitter platform. However, this limitation hinders these models from effectively capturing the intricate nuances of the Vietnamese social media context. The primary reason is that Twitter isn't a widely utilized social media platform in Vietnam, and therefore, the training data from this source may not accurately encapsulate the linguistic intricacies and contextual subtleties prevalent within the Vietnamese social media landscape. See the table below for the full results.
>
> | **Model**      | **Avg** | **Emotion Recognition** |          |          | **Hate Speech Detection** |          |          | **Sentiment Analysis** |        |        | **Spam Reviews Detection** |          |          | **Hate Speech Spans Detection** |          |          |
> | :------------- | :-----: | :---------------------: | :------: | :------: | :-----------------------: | :------: | :------: | :--------------------: | :----: | :----: | :------------------------: | :------: | :------: | :-----------------------------: | :------: | :------: |
> |                |         | Acc                     | WF1      | MF1      | Acc                       | WF1      | MF1      | Acc                    | WF1    | MF1    | Acc                        | WF1      | MF1      | Acc                             | WF1      | MF1      |
> | XLM-T          | 72\.23  | 64\.64                  | 64\.37   | 59\.86   | 86\.22                    | 86\.12   | 63\.48   | 75\.66                 | 75\.6  | 75\.6  | 90\.07                     | 90\.11   | 76\.66   | 90\.88                          | 90\.88   | 85\.53   |
> | Bernice        | 72\.49  | 64\.21                  | 64\.27   | 60\.68   | 86\.12                    | 86\.48   | 64\.32   | 74\.57                 | 74\.90 | 74\.90 | 90\.22                     | 90\.21   | 76\.89   | 90\.48                          | 90\.06   | 85\.67   |
> | **PLM4ViSMT** | **75\.65**  | 68\.10                | 68\.37 | 65\.88 | 88\.51                  | 88\.31 | 68\.77 | 77\.83                 | 77\.75 | 77\.75 | 90\.99                   | 90\.92 | 79\.06 | 91\.62                        | 91\.57 | 86\.80 |
>
> Since Bernice is the only multilingual model trained from scratch on Twitter data with a custom tokenizer, we also conducted experiments to verify its tokenizer on Vietnamese social media text. Unsurprisingly, our experiments revealed that Bernice's tokenizer performs poorly compared to most of our established baseline models, excluding viBERT and mBERT. This outcome further emphasizes the significance of our PLM4ViSMT's specialized tokenizer in effectively handling Vietnamese social media text (which is showcased in Section 3.3, "The Vietnamese Social Media Tokenizer" of the paper). See the table below for the results.
>
> | **Model**      | **Avg** | **Emotion Recognition** | **Hate Speech Detection** | **Sentiment Analysis** | **Spam Reviews Detection** | Hate Speech Spans Detection  |
> | :------------- | :-----: | :---------------------: | :-----------------------: | :--------------------: | :------------------------: | :------------------------------: |
> | XLM-T          | 26\.77  | 18\.59                  | 17\.74                    | 39\.45                 | 39\.21                     | 18\.87                           |
> | Bernice        | 28\.41  | 19\.87                  | 18\.82                    | 41\.41                 | 41\.74                     | 20\.20                           |
> | **PLM4ViSMT** | **23\.98**  | 16\.23                  | 15\.30                    | 37\.30                 | 34\.81                     | 16\.26                           |
>
> Upon publication, we will move these results forward to the main experimental results using the extra page of the camera-ready version.
>
> **References**
>
> [1] Barbieri et al., "XLM-T: Multilingual Language Models in Twitter for Sentiment Analysis and Beyond" LREC 2022
>
> [2] DeLucia et al., "Bernice: A Multilingual Pre-trained Encoder for Twitter" EMNLP 2022
>
> **Q: Line 215: large corpus of 01 GB of uncompressed texts. Is this a typo?**
>
> We thank you for pointing this typo out. We noted that our pretraining data contains only 1GB of uncompressed text. We have revised it.
>
> We believe these changes improved the quality of the article and would like to thank the reviewers again for their valuable comments and suggestions.

---

### Meta-Review · Area_Chair_Ehqn · 2023-09-13

**Recommendation:** 4

**Metareview:**

This paper trains a monolingual MLM (XLM-R architecture) from scratch specifically for Vietnamese social media text. The data was scraped from popular Vietnamese social media sites and results on downstream tasks show that it outperforms other available models.

The reviewers agree that the paper, data and models can be impactful for Vietnamese social media language understanding. The authors have provided some missing experimental results during the response period while other concerns regarding further training existing models on the newly collected data have been punted to future work.

---

### Decision · Program_Chairs · 2023-10-07

**Decision:**

Accept-Main

**Comment:**

This paper trains a monolingual MLM (XLM-R architecture) from scratch specifically for Vietnamese social media text. The data was scraped from popular Vietnamese social media sites and results on downstream tasks show that it outperforms other available models.

The reviewers agree that the paper, data and models can be impactful for Vietnamese social media language understanding. The authors have provided some missing experimental results during the response period while other concerns regarding further training existing models on the newly collected data have been punted to future work.